# Current and Prospective Methods for Assessing Anti-Tumor Immunity in Colorectal Cancer

**DOI:** 10.3390/ijms22094802

**Published:** 2021-04-30

**Authors:** Yulia I. Nussbaum, Yariswamy Manjunath, Kanve N. Suvilesh, Wesley C. Warren, Chi-Ren Shyu, Jussuf T. Kaifi, Matthew A. Ciorba, Jonathan B. Mitchem

**Affiliations:** 1Institute for Data Science and Informatics, University of Missouri, Columbia, MO 65201, USA; iid49@mail.missouri.edu (Y.I.N.); shyuc@missouri.edu (C.-R.S.); kaifij@health.missouri.edu (J.T.K.); 2Department of Surgery, Columbia, MO 65212, USA; yariswamym@health.missouri.edu (Y.M.); suvileshk@missouri.edu (K.N.S.); warrenwc@missouri.edu (W.C.W.); 3Harry S. Truman Memorial Veterans’ Hospital, Columbia, MO 65201, USA; 4Bond Life Sciences Center, University of Missouri, Columbia, MO 65211, USA; 5Siteman Cancer Center, Washington University School of Medicine, St. Louis, MO 63110, USA; mciorba@wustl.edu; 6Division of Gastroenterology, Department of Medicine, Washington School of Medicine, St. Louis, MO 63110, USA

**Keywords:** colorectal cancer, anti-tumor immunity, bioinformatics, tumor microenvironment, immune surveillance

## Abstract

Colorectal cancer (CRC) remains one of the deadliest malignancies worldwide despite recent progress in treatment strategies. Though immune checkpoint inhibition has proven effective for a number of other tumors, it offers benefits in only a small group of CRC patients with high microsatellite instability. In general, heterogenous cell groups in the tumor microenvironment are considered as the major barrier for unveiling the causes of low immune response. Therefore, deconvolution of cellular components in highly heterogeneous microenvironments is crucial for understanding the immune contexture of cancer. In this review, we assimilate current knowledge and recent studies examining anti-tumor immunity in CRC. We also discuss the utilization of novel immune contexture assessment methods that have not been used in CRC research to date.

## 1. Introduction

Colorectal cancer (CRC) is a leading cause of cancer-related death among men and women worldwide and the second leading cause of cancer death in the United States [1,2]. Patients diagnosed with metastatic disease, approximately 40% of patients, have only a 14% 5-year overall survival despite recent improvements in therapy [1]. Thus, there remains a critical need for improved therapy and understanding of therapeutic resistance in CRC, particularly for metastatic disease. Immune-based therapy, particularly in the form of immune checkpoint inhibition, has dramatically improved survival in a number of difficult-to-treat malignancies, such as melanoma, non-small cell lung cancer, and renal cell cancer [3]. However, due to the high inter-patient and intratumor heterogeneity of CRC, immune therapy is effective in only a small minority of patients. In CRC, immune checkpoint inhibition therapy has proven to be effective primarily in patients with tumors exhibiting high microsatellite instability (MSI-H) [4,5]. These tumors are characterized as having high tumor mutational burden, increased tumor-associated neoantigens, and increased tumor-infiltrating lymphocytes (TILs), which is thought to be the reason for the observed immune responses [6,7]. CRC treatment requires controlling tumor cell growth as well as activating the tumor microenvironment (TME) to promote anti-tumor immunity (Figure 1a). The immune system responds to tumors in both positive and negative ways (Figure 1b,c). Therefore, deconvolution of cellular components in the highly heterogenous TME is crucial for understanding anti-tumor immunity in cancer. Previous studies have emphasized that different subsets of tumor-infiltrating immune cells (TIICs) are correlated with cancer development and progression. However, different experimental approaches have led to inconsistent results in terms of TIICs contribution to CRC clinical outcomes [7,8,9,10]. In-depth assessment of patients’ tumors using next-generation sequencing (NGS) technology, access to publicly available data, and emerging biomedical informatics approaches have brought into question some of the theories proposed based on conventional methods of tumor immunology research [11,12,13,14,15,16]. Additionally, high-throughput genomic technologies such as tumor microenvironment cell estimation methods, single-cell sequencing, and spatial transcriptomics enable the emergence of new domains in onco-immunology that enhance our understanding of CRC [14,17,18,19].

Considering its therapeutic promise for a subset of CRC patients and numerous other malignancies, immunotherapy represents a major area of CRC treatment development. However, in many patients, the molecular basis of immunotherapy resistance remains elusive. In general, heterogenous cell groups in the TME are considered the major barrier for unveiling the causes of low immune response. Advances in technologies that can be utilized for immune profiling to the single-cell level have the potential to revolutionize the understanding of the mechanisms of anti-tumor immune response and clarify causes of heterogeneous responses to immunotherapy. In this review, we consolidate knowledge about current and prospective methods for the study of anti-tumor immunity in CRC.

## 2. Conventional Methods of Immune Contexture Estimation in CRC

### 2.1. Histology-Based Methods

The historically predominant method for the quantification of immune cell infiltration in cancer has been by histopathologic evaluation of tumor tissue. These methods have primarily been used to obtain characteristics such as size, histological grade, depth of invasion, tissue integrity, evidence of proliferation, and lymphovascular invasion. This standard evaluation is utilized as a clarification system for tumor staging and is generally focused on the tumor cells rather than the effect of the host immune response. However, histopathological examination of hematoxylin and eosin (H&E) stained formalin-fixed paraffin-embedded (FFPE) sections does enable detection of lymphoid immune infiltration. The first assessment of tumor-infiltrating lymphocytes (TILs) in rectal cancer with a semi-quantitative H&E-based scoring system was published in 1986 [22]. Using this method, survival was predicted based on the level of lymphocytic infiltrate, which was scored as little/none, moderate, or pronounced. Another scoring method based on a semi-quantitative H&E assessment is the Klintrup–Makinen (KM) score [23]. The inflammatory cell reaction was estimated in H&E stained sections in central areas of each tumor and at the invasive margin. The resulting classification of inflammatory cell response had prognostic significance [24,25].

Immunohistochemical (IHC) analysis is another image-based analysis; however, this allows for a more detailed characterization of TIICs. More specifically, IHC is an antibody-based system that detects and marks certain subsets of immune cells in the tumor microenvironment (TME). Cells commonly identified to establish the contexture of the tumor environment are typically the subject of this analysis, such as tumor-infiltrating T cells (CD3+), helper T cells (Th, CD4+), cytotoxic T-cells (Tc, CD8+), memory T cells (Tm, CD45RO+), macrophages (TAM, CD68+), and PD-1+ T-cells, among others. Some of the earliest studies demonstrating the importance of T cell infiltration on disease recurrence and survival were on colorectal cancer, revealing the prognostic value of CD3+ T-cell infiltration on progression [26,27]. Further, the same group expanded on this method to quantify Tc and Tm cells in the tumor microenvironment. They not only found a significant correlation between the infiltration of these specific cells but also that the location of the cells in the tumor (center or invasive margin) predicted patient survival and recurrence after surgical resection [27]. The marriage of this technique with digital pathology then led to the development of the Immunoscore, which is based on the quantification of CD3 and CD8 positive T cells, their density, and their location [7]. The Immunoscore has demonstrated significant prognostic value validated in multiple studies internationally and has been suggested as an adjunct to the tumor–node–metastasis system, although it is not currently used widely in the clinic [28]. Building on the techniques of IHC is another method known as immunofluorescence (IFC). Immunofluorescence, as suggested in the name, uses fluorescent-labeled antibodies in a similar way as IHC; however, images must be acquired using a fluorescent microscope [29]. Using this method and specific equipment, such as a confocal microscope, cells can be identified by the expression of multiple markers (Figure 2) [30,31]. Additionally, as in IHC, the location of cells can be marked in the TME, as well as information such as proximity to vascular structures [32]. This method also has been used for ex vivo profiling of tumor-derived organoids, as known as tumoroids, to establish the relevant architecture and expression of cell surface molecules [33]. In these systems, IFC can also be used to measure cell proliferation and death [33,34].

A significant limitation of the above techniques of IHC and IFC is that they are unable to look at more than a few markers on each slide. In response to increasing demand to simultaneously detect multiple markers from one tissue section, multiplexed IHC/IFC (mIHC/IFC) techniques have been developed and adopted in both research and clinical settings [35]. Generally, mIHC utilizes chromogenic and fluorogenic techniques. In chromogenic mIHC, markers are identified using antibodies raised in the same or different species that are directly labeled with different chromogens [36,37]. The number of markers is limited to four due to a narrow visible spectrum. Fluorogenic mIHC uses tyramide signal amplification, which covalently labels the tissue section with fluorescent immunostains for each marker. The process is repeated through several rounds of antibody stripping. Fluorogenic mIHC enables the detection of up to six markers in a tissue section [38]. The mIHC/IF is widely used in cancer immunotherapy research. Thus, in a recent review of the methods for prediction of clinical response to anti-PD-1/PD-L1 therapy, mIHC/IF method had significantly higher diagnostic accuracy than PD-L1 IHC, tumor mutational burden, or gene expression profiling [39]. This method has become particularly useful for CRC research and immune profiling [40,41,42,43,44].

### 2.2. Cytometry-Based Methods

Another technique that allows for the measurement of cell characteristics, such as size and granularity, as well the expression of specific proteins, is fluorescence-activated cell sorting (FACS) or flow cytometry (FC). In this technique, similar to IFC, cells are labeled with monoclonal antibodies bonded to fluorescent molecules [45,46]. Using microfluidics, cells are then organized singularly in a stream and exposed to different wavelengths of light to measure the expression of specific markers in a single-cell manner. Given the ability to measure the characteristics of individual cells in the TME, this is an excellent modality for determining specific tumor-associated populations [47,48]. Scurr et al. used flow cytometry to analyze Tregs in CRC and revealed that tumors contained more highly immunosuppressive CD4+FOXP3- Treg-like cells compared to normal colon tissue or blood [49]. Additionally, they were able to demonstrate that this population of cells expressed CD39, cytotoxic T-lymphocyte antigen (CTLA-4), and produced IL-10 and TGF-beta, a key finding in demonstrating the importance of these cells in CRC. Other groups have utilized flow cytometry to successfully identify several subsets of immune cells in colorectal cancer. Girardin et al. revealed a lower frequency of effector T cells (CD8+CD69+) but a higher frequency of both regulatory (CD25hi Foxp3+) and inflammatory helper T cells (IL-17+) compared with normal bowel tissue [50]. Another study showed that T cells in the TME produced more IL-17 and less IL-2 per cell than T cells from non-tumor-bearing tissue [51]. Building on the ability to identify single cells, FACS can also be utilized to separate out specific cells in a sterile manner for further experiments. This type of analysis has been critical for the identification and delineation of the characteristics of cells such as myeloid-derived suppressor cells (MDSC) in CRC [52]. However, the overall power of FC and FC-associated cell sorting is limited by the overlapping emission spectra of the fluorochromes [53].

Mass cytometry circumvents the limitations of flow cytometry with the concept of labeling cells with heavy metal isotopes instead of fluorescent labels. Also known as cytometry by time-of-flight (CyTOF), this method combines the high throughput of flow cytometry and the fine resolution of mass spectrometry. This attribute limits interaction between overlapping light emission spectra and significantly expands the number of markers that can be measured, as the only limitation is the number of heavy metals that can be used. Utilized in immunoprofiling of CRC, CyTOF was utilized to reveal that the presence of EpCAM+ CD4+ T cells might be a sign of colon cancer development [54]. Expanding on the ability to undergo high fidelity analysis, another group used CyTOF to evaluate murine colon adenocarcinomas and identify the critical cells involved in response to PD-L1 along with other immune-based therapy, such as anti-LAG3 and ICOS agonism [55]. These findings were confirmed using selective markers in human CRC with FC. This method has also been used as a way to confirm the high fidelity results associated with single-cell genomics studies we describe later, such as to characterize the immune infiltrate associated with normal and colitic colons in patients with ulcerative colitis [56]. However, CyTOF is limited by the method in some ways. The cells are necessarily destroyed during the process of measuring the markers, so there is no option for downstream functional studies as they can be completed with FC associated sorting methods. As with single-cell-based genomic techniques, CyTOF also requires significant downstream data wrangling but is a useful method for biologic studies, particularly when paired with other methods.

## 3. Transcriptome Analysis Using Next-Generation Sequencing

The emergence of NGS technologies along with comprehensive and coordinated efforts to obtain human tissue samples (e.g., The Cancer Genome Atlas (TCGA)) enabled the characterization of multidimensional maps of genomic changes in common cancers [57]. Availability of such comprehensive information opened the era of data-driven bioinformatics tools [58]. Particularly, in combination with knowledge regarding cancer immunity, this combination of data and tools has opened the door to greater study of tumor-immune cell interaction mechanisms. Fakih et al. discovered a subcohort of patients with high CD8+ T cell infiltration and poor clinical outcomes re-analyzing CRC patients’ data from TCGA [59,60].

The first mathematical methods of cell type quantification from transcriptomic data started appearing at the beginning of this century [61]. Gene Set Enrichment Analysis (GSEA) was used to score immune cell subsets in heterogeneous samples based on previously established transcriptomic signatures for certain subclasses of immune cells both in normal tissue and cancer [62,63]. Using these techniques, Angelova et al. developed 31 custom gene sets for TIL assessment in CRC patients [64]. With this same approach, Charoentong et al. established 28 pan-cancer immune sets for 10 different solid tumors [65]. Next, deconvolutional methods for TILs quantification began to emerge. Estimation of STromal and Immune cells in MAlignant Tumours using Expression data’ (ESTIMATE) uses single sample GSEA (ssGSEA) to calculate immune and stromal scores to predict the levels of immune and stromal cells infiltration and infer tumor purity [66]. Using ESTIMATE on the CRC progression dataset, Liu et al. showed that decreased levels of immune scores in primary and metastatic CRC compared to a normal colon correlate with cancer progression [67].

Starting from linear regression-based heuristic algorithms, highly accurate tools for identifying cell subpopulations in the TME were developed [68,69,70,71]. The Tumor Immune Estimation Resource (TIMER) web-server allows to comprehensively investigate molecular characterization of tumor–immune interactions. In their original paper, Li et al. calculated levels of six tumor-infiltrating immune subsets for 10,897 tumors from 32 cancer types, forming a basis for estimation of tumor-infiltrating immune cells’ abundance. To predict the abundance score, they apply a constrained least-squares fitting on the expression of the genes that are negatively correlated with tumor purity [72]. In 2020, the authors published TIMER version 2.0, which uses six state-of-the-art algorithms for more robust estimation of immune infiltrating levels. Additionally, there are modules for exploration of the associations between the levels of immune infiltrates and genetic or clinical features as well as cancer-related associations in the TCGA cohorts [73].

Cell-type Identification by Estimating Relative Subsets of RNA Transcripts (CIBERSORT) uses support vector regression and allows for the estimation of the relative proportions of 22 immune cell subtypes within heterogeneous tumor samples [74]. Using CIBERSORT as a tool for TILs quantification in TCGA CRC cohorts, Zhao et al. confirmed immunoscore independent prognostic values [75]. A different team, using similar methods, revealed that a high density of infiltrating tumor-associated neutrophils (TANs) was associated with better prognosis in CRC patients while a high number of Tregs and tumor-associated macrophages (TAMs) had shorter disease-free and overall survival [76]. Although the CIBERSORT method is useful for intratumor heterogeneity assessment, it has limitations in intertumor comparisons. This method relies on the relative quantity of specific mRNA transcripts so it provides an estimate of the relative percent of specific cell types within a tumor, but this measure is not readily extended to compare cellular composition between patients [77]. In contrast, the Microenvironment Cell Populations counter (MCP-counter), a methodology also based on gene expression, has demonstrated greater utility for quantifying cell subpopulations proportionally to the amount of cells within a tumor, allowing inter-sample comparison [78]. MCP-counter has been being used in exploratory analyses of consensus molecular subtypes (CMSs) in CRC [79,80,81]. This method was used to demonstrate that the CMS1 subgroup was highly enriched in cytotoxic T cells and had high immune checkpoint expression along with an IFNγ signature, high class I major histocompatibility complex (MHCI) antigen expression, moderate inflammation, and angiogenesis, consistent with the initial findings of CMS grouping authors [82]. Additionally, further assessment of the TME using these methods revealed the absence of IFNγ and high inflammatory, angiogenic, and fibroblastic invasion in CMS4 patients, shedding light on the poor response to immune therapy in these patients [77]. Another study utilizing the MCP-counter showed that TP53 mutation in addition to a CMS profile has immunobiological associations with prognostic and potentially immunotherapeutic implications [83]. Becht et al. utilized the MCP-counter and reported that the MSI-like CMS1 subgroup contained higher densities of CD8+ and CD68+ cells compared to canonical CMS2 and metabolic CMS3 subtypes with intermediate prognosis, which exhibit low immune and inflammatory signatures. The presence of cytotoxic T cells and low expression of fibroblast-related genes, which is associated with low presence of cancer-associated fibroblasts (CAFs), is correlated with good prognosis and, therefore, with the most valuable outcome for the patients [83]. Using the MCP-counter along with pathway enrichment analysis, Shen et al. demonstrated significant differences in cytotoxic lymphocyte (CTL) invasion in CRC patients based on the location of the tumor and the stage and found conserved pathways of immune dysregulation associated with survival [6]. Additionally, they found that patients with CTL deficient right-sided metastatic CRC had the most pathways associated with survival; particularly important as these are the patients with the poorest survival in all subgroups.

Due to the availability of bulk transcriptomic data and cost-effectiveness of analysis, the methods described above remain prevalent in anti-tumor immunity assessment. However, those methods are limited in some cases related to the detection of rare or unknown cell types. Moreover, mRNA expression does not always directly correlate to protein levels [84,85,86]. Newer methods focused on single-cell technology attempt to improve on these limitations, which we will discuss below.

## 4. Analysis of TME on Single-Cell Level

### 4.1. Single-Cell Transcriptomic, Genomic, and Proteomic Analysis

Previously described FC and IHC methods have been used for immune profiling for many years. Recently, however, these techniques have been expanded in combination with a number of NGS-based technologies such as single-cell transcriptomics, genome or epigenome sequencing, and advanced analytic approaches. These newer technologies provide high-throughput methods that enable obtaining full genetic information from millions of cells, making it a natural fit for tumor heterogeneity studies and particularly studies of the immune microenvironment in cancer.

Computational analysis of intratumoral heterogeneity in bulk samples is based on the inference of subclonal structure through analysis of mutant allele frequencies. However, it is impossible to resolve some combinations of mutant allelic frequencies computationally from bulk genomic data. DNA analysis at the single-cell level may reveal the clonal structure and the order of genetic alterations, tracing dynamic clonal evolution and providing insights into critical steps in oncogenesis. Thus, in one of the first single-cell sequencing studies, Yu et al. showed that colon cancer could be of a biclonal origin. They also suggested that mutations in the SLC12A5 gene that were not seen in bulk colon cancer sequencing data may be a cancer driver [87]. This demonstrates the power of single-cell genomics discovery as compared to bulk sequencing techniques.

Single-cell transcriptome sequencing (scRNA-seq) may lead to further resolution of major mysteries in intratumoral heterogeneity, including deconvolution of cell populations in the TME, trajectory inference of cell fates, and discovery of rare cell types in cancer ecosystems. In 2011, single-cell qPCR analysis provided by Dalerba et al. demonstrated that colon cancer cell subpopulations mirror normal transcriptional identities of different cellular lineages in the normal colon. This finding suggested that in vivo lineage differentiation could be the major reason for intratumoral heterogeneity, at least in colon cancer [88]. Further, scRNA-seq analysis of different cancer types revealed previously unknown cell subpopulations that may contribute to tumorigenesis, tumor progression, and other processes. Moreover, the ability to assess gene expression within those subpopulations allows a more detailed reconstruction of mechanisms behind cancer initiation and progression. Thus, Li et al. developed reference component analysis (RCA) and found two distinct subtypes of cancer-associated fibroblasts (CAFs) with high expression of epithelial-to-mesenchymal transition (EMT) marker genes. Having analyzed gene signatures for newly recognized cell subtypes and clinical outcomes in the TCGA, they suggested that CRC tumors previously assigned to the enterocyte and goblet-like classes could be divided into two subgroups with different survival outcomes [18].

Besides discovering rare cell types, scRNA-seq provides further opportunity to investigate immune cell subpopulations within the TME. These data can then be used to illuminate anti-tumor immunity and to determine specific cellular populations that associate with a clinical response to checkpoint blockade and other immune-based therapy. The standard pipeline of scRNA sequencing and analysis is shown in Figure 3. Here, we demonstrate cell preparation and capturing based on 10X Genomics technology, which is used widely in scRNA-seq research. Single cells, reverse transcription (RT) reagents, gel beads containing barcoded oligonucleotides, and oil are combined on a microfluidic chip. Cells are captured by the beads and form reaction vesicles called Gel Beads in Emulsion (GEMs). After RT, cDNAs from a single cell will have the same barcode, so the sequencing reads will be able to be mapped back to their original single cell. Raw sequencing data are then processed to count matrices where each cell has gene expression data. Quality control allows the removal of potential doublets, empty cells, and cells with high expression of mitochondrial genes as they might be apoptotic or lysed. The normalization step addresses the issue of discrepancies in counts of identical cells due to sampling effects. Usually, a priority list of up to 5000 “highly informative” variable genes is used for downstream analysis [89]. Dimensionality reduction is usually performed using principal component analysis (PCA) to use the most significant genes. Unsupervised clustering defines cell populations with similar gene expression patterns [90]. Downstream analysis may include but is not limited to trajectory inference and differential analysis of each cluster. Various methods for each step of this pipeline have been developed rapidly during the last few years with continuous methodologic improvement.

Nevertheless, the major challenges of scRNA-seq still remain the same. The most criticized limitation of single-cell sequencing technology is that genes with low expression tend to be dropped or are susceptible to technological noise even if they are captured. Sometimes those lowly expressed genes can be critical cell surface markers. The second challenge is proper identification of cells clustered together. Current methods of cell type identification are mainly based on determining differentially expressed genes for each cluster and curating cell types according to the cell surface markers that were found using conventional immune methods described above [91]. However, cells may have more complicated expression patterns that distinguish them from other subtypes. Finally, single-cell sequencing data does not preserve spatial information that can be important for interpretation of immune cell function.

For immune profiling using single-cell sequencing, T cell receptor (TCR) sequencing overcomes some of the limitations discussed above. T cells are crucial for the anti-tumor immune response. They can detect an infinite variety of self as well as non-self cells. The ability of T cells to recognize so many pathogens is enabled by highly heterogeneous surface receptors called the TCR. Infinitely diverse combinations of gene segments in alpha and beta protein chains in the TCR cause a great variety of T cell subtypes that affect intratumor heterogeneity and, therefore, response to immunotherapy in cancer. Additionally, subsequent heterodimeric alpha and beta chain pairing increases the number of possible combinations [92,93]. Before the single-cell era, the TCR had been sequenced using “bulk” sequencing technologies, which could not account for alpha and beta chain pairings [94,95,96].

Single-cell sequencing overcomes that limitation and allows the researcher to distinguish T cell subtypes by getting gene expression information and the TCR repertoire from each cell [97]. All single-cell protocols include reverse transcription and amplification before library preparation. The resulting cDNAs can be used for TCR sequencing for each cell along with their expression information. TraCer was an early tool to reconstruct alpha and beta chain pairing. In their experiments, Stubbington et al. successfully applied TraCer to show CD4+ T cell subset dynamics upon Salmonella infection [98]. Subsequently, Zhang et al. presented STARTRAC, a framework that accounts for T cell cluster distribution and migration across tissues, clonal expansion, and transition between developmental states. STARTRAC uses scRNA-seq and TCR sequences from peripheral blood, tumor, and adjacent normal tissues [14]. In the same work, Zhang et al. applied STARTRAC to CRC patients’ samples containing 11,138 cells from the tumor, adjacent normal tissues, and peripheral blood. As a result of clustering, 12 CD4+ and 8 CD8+ T cell clusters were found, including CRC-specific T cell subtypes such as Th17, follicular T helper cells, follicular T regulatory cells. Among CD8+ cells, they identified subsets associated with exhaustion (Tex), effector memory (Tem), recently activated effector memory (Temra), suggesting specific patterns that may be targetable for improving immune therapy. Comparing MSI and MSS patients’ cell content, the authors found that only CD4+ CXCL13+BHLHE40+ TH-like cell clusters were significantly enriched in patients with MSI. IGFLR1, a previously uncharacterized gene was upregulated in that cluster as well as in CD8+ exhausted T cells. Having provided an additional in vitro experiment, Zhang et al. suggested that IGFLR1 may be a co-stimulator in TCR signaling. Based on these findings, the authors developed the iSTARTRAC web platform with the TIL dataset [99]. In 2020, the same group utilized two single-cell sequencing platforms with different sensitivity and throughput to analyze 54,385 cells from 18 CRC treatment-naïve patients. Two distinct subpopulations of myeloid cells with different responses to CSF1 blockade were found. Additionally, anti-CD40 treatment activated conventional dendritic cells (cDCs) and increased Bhlhe40+ Th1-like cells and CD8+ memory T cells [100]. This demonstrates the power of these analyses to identify and alter treatment.

Besides the application of TCR sequence assembly to scRNA-seq data, there are other tools for trajectory inference that use single-cell expression data [101,102]. Masuda et al. used Monocle in their analysis of scRNA-seq, TCR sequence, and 23 surface proteins of 37,931 single T cells from CRC patients. They found a CD38+ peripherally-derived regulatory T cell subset that was correlated with poor clinical prognosis [103]. In their comprehensive single-cell study of CRC molecular subtypes, Lee et al. showed that tumors with different molecular signatures have unique immune microenvironments. Having provided clustering, trajectory inference, and cell–cell interaction analysis, they suggested that unique immune ecosystems might be affected by BRAF/KRAS mutations, which were more prevalent in non-CMS2-like tumor cells and SMAD4 mutations that support tumor cell survival in a TGF-beta-rich microenvironment formulated by myeloid cells and myofibroblasts [104].

Although sc-RNA-seq provides excellent data for the reconstruction of high-resolution cellular maps and discovering new cell subpopulations, it suffers from the caveat that mRNA and protein levels do not always correlate. To detect mRNA and proteins from each cell at the same time, new technologies were developed based on antibody-tagged oligonucleotides. In 2017, cellular indexing of transcriptomes and epitopes by sequencing (CITE-seq) was introduced by Stoeckius et al. [105]. The method involves the conjugation of antibodies to the 5′ end of oligos through streptavidin–biotin interactions. Subsequently, the cells are immunostained by antibody-oligo complexes using flow cytometry protocols. Following cell lysis, cellular mRNA and antibody-derived oligos anneal via their polyA tails to the polydT tail of the barcoding beads. During reverse transcription, both the mRNA and antibody-derived oligos are indexed by a shared cellular barcode. The resulting transcriptomic- and proteomic-derived material can then be separated by size and converted into Illumina-based sequencing independently. The proof of concept demonstrated simultaneous detection of 13 monoclonal antibodies.

Other single-cell, multi-omic approaches have also been developed to address this issue. Similar to CITE-seq, RNA expression and protein sequencing assay (REAP-seq) was used to quantify 82 antibodies and more than 20,000 genes [106]. The main difference from CITE-seq is that REAP-seq uses amine chemistry for conjugation of oligonucleotides to antibodies. Another single-cell protein profiling method called Abseq was also introduced in 2017 [107]. The concept of the technology is the same as in CITE-seq and REAP-seq. However, Abseq focuses on the detection of single-cell protein levels and no mRNA. Furthermore, Abseq uses a highly advanced custom microfluidic workflow that consists of three devices instead of one for CITE-seq and REAP-seq. That adds significant technological challenges. Nevertheless, all three methods described above are able to fill the gap in the discrepancy between mRNA levels and protein amounts in scRNA-seq. Thus, in the study profiling myeloid cells in glioblastoma, CITE-seq revealed new cell markers for subsets of TAMs and dendritic cells (DCs) that were not identified in scRNA-seq analysis [108]. Novel extensions of CITE-seq, such as ECCITE-seq, that integrates pooled CRISPR screens into CITE-seq measurements and Perturb-CITE-seq, which combines pooled genetic perturbation screens with CITE-seq, were used to investigate molecular mechanisms underlying cancer immunotherapy resistance [109,110,111]. Although, to the best of our knowledge, the methods described above have not been being applied in CRC yet, CITE-seq or its modifications can expand our knowledge of intratumor heterogeneity and immune evasion in this deadly disease.

### 4.2. Single-Cell ATAC-seq

As described above, one of the limitations of using scRNA-seq for cell type identification is using cell surface markers that were discovered by non-single-cell methods. The single-cell Assay for Transposase-Accessible Chromatin using sequencing (scATAC-seq) allows marker-free identification of cell types using regulatory elements. Expression of genes that are specific for particular cell types is affected by cis-acting DNA elements and trans-acting factors. The scATAC-seq method uses mutant Tn5 transposase to cut and insert assessable chromatin adapters into the genome. After PCR amplification with barcoding, the libraries undergo high-throughput sequencing, which outputs multidimensional assays of the regulatory landscape of chromatin [112]. Raw reads are then mapped to a reference genome and the resulting files undergo peak calling and read counting. scATAC-seq downstream analysis uses a matrix with a number of reads on peaks for each cell. Dimension reduction for a scATAC-seq sparse matrix is usually done using PCA, as in scRNA-seq analysis. Latent semantic indexing (LSI) methods initially created for natural language processing are also commonly used for dimension reduction in scATAC-seq [113].

Similar to scRNA-seq data analysis, UMAP or tSNE non-linear dimension reduction for visualization of clusters is utilized. Differential peak calling is performed using non-parametric methods such as the Mann–Whitney U test or Wilcoxon rank-sum test for calculation of the peaks specific for the cluster of interest versus all the cells outside that cluster. There are different methods that are used for cell type annotation that aim to find whether a particular TF motif is enriched in each cell [114]. Another way to identify cells is finding significant binding overlaps between differential scATAC-seq peaks and public ChIP-seq data [115]. Although it has the ability to precisely decipher intracellular heterogeneity without using cell markers, scATAC-seq has been less frequently adopted in cancer research because of its relatively high cost. Satpathy et al. used scATAC-seq to analyze the chromatin profile before and after treatment with anti-PD1 immunotherapy in patients with basal cell carcinoma. They found increased CD8+ exhausted T cells and T follicular helper cells post-treatment. In addition, both cell types had shared transcription factors along their differentiation following immunotherapy [116]. Used as a part of a multiomics study, scATAC-seq analysis uncovered an ibrutinib-induced regulatory program in chronic lymphocytic leukemia (CLL). Chromatin accessibility over the course of inbrutinib treatment was measured and the results showed NF-kB mediated B cell identity in CLL. Non-malignant immune cells’ chromatin accessibility changed in cell-type-specific ways during the treatment sharing a quiescence-like gene signature [117]. At present, this method has been applied primarily in non-CRC malignancies, leaving this open for future studies in this patient sub-group.

### 4.3. Spatial Transcriptomics

While technologies for obtaining single-cell transcriptomic and epigenomic data analysis provide a fine-grained picture of the TME structure, they are not able to preserve location information of cells inside tissues. On the other hand, there are IHC and FISH methods that allow for determination of the location of cells expressing certain markers, but they are limited by the number and type of markers available. To combine high throughput sequencing with mapping cell location, the spatial transcriptomics (ST) method has been developed. Using this method, location information for gene expression data is preserved in tissue by immobilizing barcoded oligonucleotides on a glass slide and mounting tissue on it. After permeabilization and reverse transcription of mRNA from the tissue, the probe is removed, and cDNA was sequenced. The results show gene expression at their original location in the tissue. Despite its advantage of uncovering the precise location of sequences of interest, ST was not resolved at the single-cell level as each spot covered several cells [118].

Slide-seq is another technology for genome-wide readout of gene expression with spatial data. Using DNA-barcoded beads, Slide-seq allows to mark and sequence larger tissue sections at higher resolution. They developed non-negative matrix factorization regression (NMFreg) to reconstruct the expression of each bead as a weighted combination of cell type signatures defined by scRNA-seq [119]. High-Definition Spatial Transcriptomics (HDST) uses a pooling method to produce barcoded beads of 2 mm size for which the position is decoded by a sequential hybridization and error-correcting strategy. HDST demonstrated 25 times higher resolution than Slide-seq in one study [120].

High throughput spatial technologies are used for immune profiling less often than scRNA-seq, which is less expensive and more accurate. Currently, spatial mRNA-seq methods are used as an additional source of information for scRNA-seq. Thus, ST was used for analysis of cellular composition in cutaneous squamous cell carcinoma (cSCC). The combination of scRNA-seq and ST revealed that tumor-specific keratinocytes (TSKs) act as a hub for intercellular communication. The study showed that physical proximity of specific cell types such as TSK, basal, and adjacent stromal and immune cells is associated with their invasiveness and immunosuppressive characteristics [121]. Again, this method has not been widely applied to CRC, and there is room for utilization of this technique in these patients.

## 5. Conclusions

In recent years, immunotherapy has achieved impressive results in treating various cancer types, including a small subset of CRC patients. Despite dramatic progress in onco-immunology research, the underlying mechanisms of immunotherapy resistance in the majority of CRC patients remains elusive. Precise profiling of the immune contexture of the CRC TME will help to further reveal the underpinnings of anti-tumor immune mechanisms. Conventional methods of profiling the TME are limited in-depth and therefore do not show a detailed and scaled picture. Next-generation sequencing and big data technologies have greatly expanded our ability to evaluate the immune landscape of cancers enabling the study of immune infiltration on a high throughput level. Further development of techniques, such as scRNA-seq, have revealed rare cell subtypes, including immune cell subpopulations with different responses to therapy [15,103]. scRNA-seq was also successfully used to reconstruct intercellular networks among tumor and immune cells in CRC [107]. Although scRNA-seq technology enables obtaining fine-grained information about the TME, the main critiques of it include using surface markers for cell identification and lack of spatial information. scATAC-seq may solve the first limitation as it uses TFs to identify cells. Additionally, spatial transcriptomics and its novel variations may allow for enhanced understanding and precise location of the novel cell types identified using scRNA-seq. With advances in single-cell-based technology and analysis methods, precise immune profiles with spatial information will reveal new information important for immune therapy insights.

## Figures and Tables

**Figure 1 ijms-22-04802-f001:**
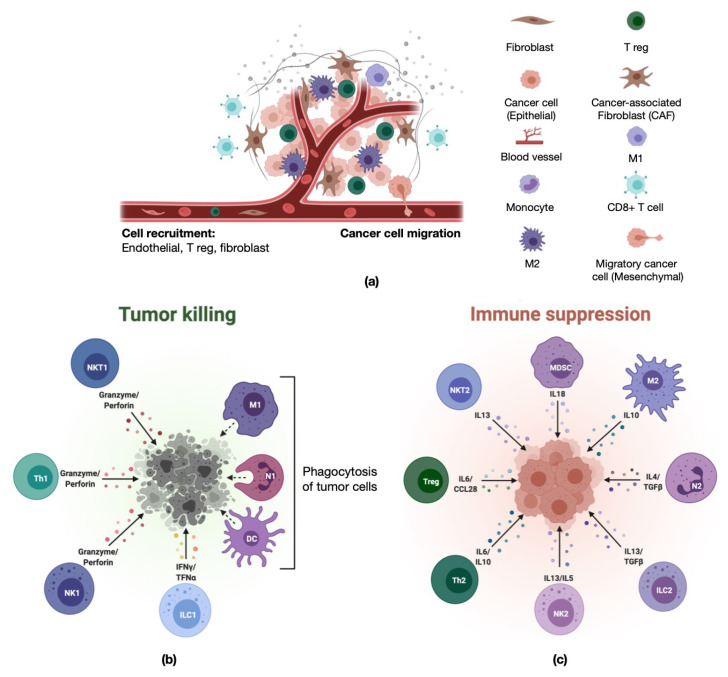
Schematic representation of the tumor microenvironment and its cellular composition. (**a**) Tumor microenvironment. Graphic representing various cellular components of vasculature and tumor microenvironment [20]. (**b**,**c**) Immune cell composition in the tumor microenvironment: cellular and molecular components involved in pro-inflammatory, tumor-killing activity (**b**) and anti-inflammatory, immunosuppressive, tumor-promoting activity (**c**) [21].

**Figure 2 ijms-22-04802-f002:**
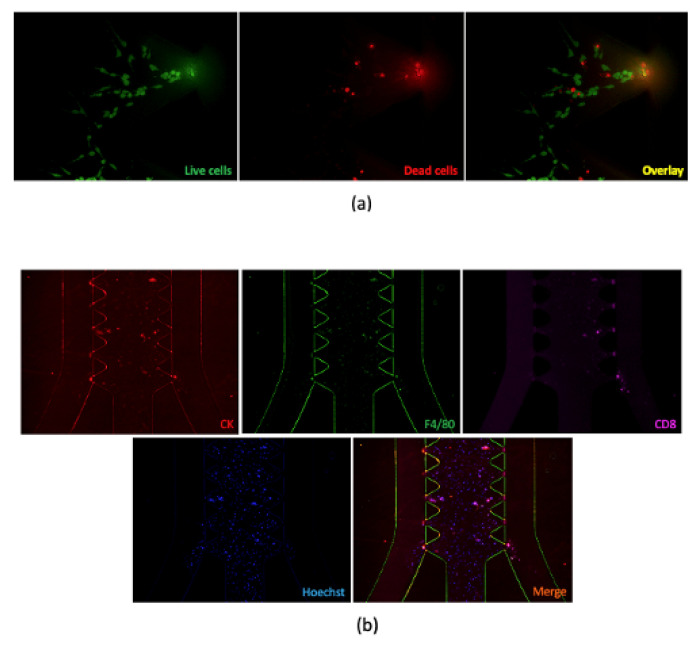
Examples of using immunofluorescence. (**a**) Live–dead cell staining for MC38 cell lines cultured in AIM 3D Chip. (**b**) Immunostaining of MC38-tumor-derived spheroids. Spheroids were stained with conjugated antibodies targeting panCK, CD4, and CD8 (overnight at 4 °C). Hoechst 33,342 was used to label nuclei. Images were taken using a fluorescence microscope.

**Figure 3 ijms-22-04802-f003:**
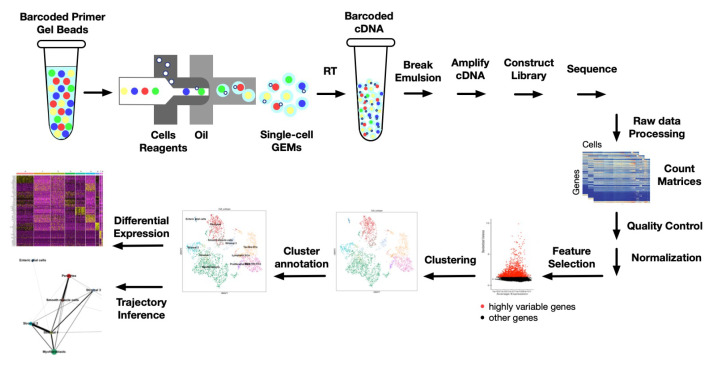
Schematic representation of scRNA sequencing and data analysis pipeline.

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
