# Peer review of "Current and Prospective Methods for Assessing Anti-Tumor Immunity in Colorectal Cancer"

_ijms, 2021, doi:10.3390/ijms22094802_

Round 1

Reviewer 1 Report

The manuscript focuses on very important issue – the colorectal cancer treatment. The manuscript reviews analyses used for the research on molecular background of antitumor immunotherapy since it is effective only in small group of colorectal cancer patients. The malignancy is still difficult to be successfully treated. The authors described the massive parallel methods of deep sequencing or analyses conducted on chips which are able to analyze whole genome or transcriptome for the one patient or one tumor cell. The manuscript is interesting. However, I have some doubts about the title of the manuscript. I would withdraw the word informatics. Nowadays molecular methods in genetics need quite sophisticated software to analyze the raw results obtained using massive methods. Especially that the authors did not describe statistical or mathematical methods which were used to analyze obtained results and they have to be done by experts in mathematics and biostatistics.

Author Response

Dear Reviewer 1:
 We would like to thank you for your review of our manuscript entitled “Biomedical
Informatics Methods for Revealing Anti-Tumor Immunity of Colorectal Cancer” by Yulia I.
Nussbaum, Yariswamy Manjunath, Kanve Suvilesh, Chi-Ren Shyu, Jussuf T. Kaifi, Wesley C.
Warren, Matthew A. Ciorba, and Jonathan B. Mitchem. We appreciate your consideration and help
in improving our work. We have answered the comments below and outlined the changes below.

1. I have some doubts about the title of the manuscript. I would withdraw the word informatics.
Nowadays molecular methods in genetics need quite sophisticated software to analyze the raw
results obtained using massive methods. Especially that the authors did not describe statistical
or mathematical methods which were used to analyze obtained results and they have to be done
by experts in mathematics and biostatistics.

We agree with the reviewer regarding the expertise and the content of our manuscript. We have
changed the title to: “Current and prospective methods for assessing anti-tumor immunity in
Colorectal Cancer.”

Reviewer 2 Report

This review is intended to cover biomedical informatics methods for revealing anti tumor immunity to colorectal cancer. It is however a mix between purely technical description of basic biological techniques and more advanced technologies in which bioinformatics analysis can be applied. It however lacks description of major techniques that have been applied to colon cancer immunity (see Qian, J., Olbrecht, S., Boeckx, B. et al. A pan-cancer blueprint of the heterogeneous tumor microenvironment revealed by single-cell profiling. Cell Res 30, 745–762 (2020). https://doi.org/10.1038/s41422-020-0355-0 for instance). It covers losely the current knowledge on colorectal cancer immunity but failed completely to comprehensively list and explain the advantages and limitations of current bioinformatics methodologies. I believe a more adequate title is given line 50 and 51.

major points:

  • what is the interest of detailing histology and cytometry_based techniques at length ( 3 pages) as they do not rely on bioinformatics for analysis?
  • figure 1 legends are too small
  • I disagree that heterogenous cell groups in the TME are the major barrier for unveiling the causes of low immune response. I believe it is patient heterogeneity first that prevents our understanding of cancer immunity.
  • Figure 1 is from mouse data. Is that relevant to the topic?
  • Table 1: what is the interest of the KM score in terms of patients survival relative to other clinical scores?
  • figure 3 is of poor quality, legends are too small and are generated in mice; What is the point of this figure? 
  • A major issue not discussed anywhere is the poor correlation between mRNA levels and protein expression. It would be useful if the authors could prove that deconvolution algorithms actually capture the real proportions of cells based on mRNA expression. Technologies able to do so (Ab-Seq, CITE-Seq etc) are not discussed.
  • Again what is the point of figure 4 unless to show that the authors are able to  generate a UMAP?
  • Table 2 is far from being exhaustive and does not bring useful informations.

Minor points:

  • line 29: responds to tumors
  • line 4: further / deepen/enhance?
  • line 87: immunofluorescence
  • line 126: TGF-beta
  • line 205: space missing
  • lines 212/213: class I not class1
  • line 336: TGF-beta
  • line 354: ref is missing
